# Spatial variation and determinants of mother and newborn skin-to-skin contact care practices in Ethiopia: A spatial and multilevel mixed-effect analysis

**Desalegn Girma[1]\*, Zinie Abita[2], Yilkal Negese[3], Gossa Fetene Abebe[1]**

1 Department of Midwifery, College of Health Science, Mizan-Tepi University, Mizan-Teferi, Ethiopia,
2 Department of Public Health, College of Health Science, Mizan-Tepi University, Mizan-Teferi, Ethiopia,
3 Department of Public Health, College of Health Science, Debre Markos University, Debre Markos, Ethiopia

\* desegir@gmail.com

**Data Availability Statement:** We declared that all the data underlying the results presented in the study are publicly available from the Harvard Dataverse: https://doi.org/10.7910/DVN/KCFWKI.

## Abstract

### Background

Skin-to-skin contact care practice is placing a naked baby on the mother's chest with no cloth separating them, in a prone position covered by a cloth or blanket. It improves the survival of newborns by preventing hypothermia, improving breastfeeding, and strengthening mother-to-child bonding. Nevertheless, it remains under-practiced in many resource-constrained settings. Therefore, the main objective of this study is to explore the spatial variation and determinants of mother and newborn skin-to-skin contact care practices in Ethiopia.

### Method

The study was done using the 2016 Ethiopian Demographic and Health Survey data. A weighted sample of 10417 mothers who gave live birth before the five-year survey was extracted for the analysis. Arc GIS version 10.3 and SaTscan version 10.0.2 were used for the spatial analysis. A multilevel mixed logistic regression model was fitted to identify factors associated with skin-to-skin contact care practices of mothers and newborns. Finally, a statistically significant association was declared at a P-value of < 0.05.

### Result

In this study, skin-to-skin contact care practice of mothers and newborns was non-random across Ethiopia with Moran's I: 0.48, p < 0.001. The most likely significant primary and secondary clusters were found in Addis Ababa (RR = 2.39, LLR = 116.80, p <0.001) and Dire Dewa and Harari (RR = 2.02, LLR = 110.45, p <0.001), respectively. In this study, place of delivery (AOR = 12.29, 95%CI:10.41, 14.54), rich wealth index (AOR = 1.29, 95% CI: 1.05,1.59), medium wealth index (AOR = 1.38, 95% CI:1.17, 1.68), having 1–3 antenatal care visits(AOR = 1.86,95% CI: 1.56, 2.29), having ≥4 antenatal care visits (AOR = 1.93,95% CI: 1.56, 2.39), initiating breastfeeding within the first hour (AOR = 1.75,95%

**Funding:** The author(s) received no specific funding for this work.

**Competing interests:** The authors have declared that no competing interests exist.

**Abbreviations:** AIC, Akaike's information criterion; ANC, Antenatal care; AOR, Adjusted odds ratio; CI, Confidence intervals; DIC, Deviance information criterion; EAs, Enumeration areas; EDHS, Ethiopian Demographic and Health Survey; ENAP, Every Newborn Action Plan; ENC, Essential newborn care; ICC, Intraclass correlation coefficient; LLR, log-likelihood ratio; LMIC, low and middle-income countries; MOR, Median odds ratio; PCV, Proportional change in variance; RR, relative risk; SCC, skin-to-skin contact care; SNNP, South nation nationality and people; WHO, world health organization.

CI:1.49,2.05) and media exposure (AOR = 1.20,95%CI 1.02,1.41) were factors associated with skin to skin contact care practice of mothers and newborns.

## Conclusion

This study concludes that the Skin-to-skin contact care practices of mother and newborn is not random in Ethiopia. Therefore, the implementation of essential newborn care packages should be regularly monitored and evaluated, particularly in the cold spot areas of skin-to-skin contact care practices. Besides, media advertising regarding the importance of Skin-to-skin contact care practices for mothers and newborns should be scaled up to increase the practices.

## Background

The first month of life, known as the neonatal period, is the riskiest time for the survival of newborns. Globally, around, 5.0 million under-five deaths were reported in 2020, of these newborns contributed to about 2.4 million of the deaths [1]. Nearly, 80 percent of the deaths were from low and middle-income countries (LMIC) [1]. Most newborns have died due to preventable causes such as preterm birth, birth asphyxia, and infections [2]. Considering the importance of the problem, in 2014, the World Health Organization (WHO) launched a platform "Every Newborn Action Plan (ENAP)" to end preventable newborn deaths and still-births and set a target for all countries to attain 10 or fewer neonatal deaths per 1000 live births by 2035 [3].

To achieve the above targets, the WHO has primarily focused on strengthening and improving the quality of care during antenatal care, skilled care at birth, and postnatal care [3]. Of this, essential newborn care (ENC) practices are simple life-saving interventions provided to all newborns during the birth and postnatal period [4]. Skin-to-skin contact care (SCC) practice is one of the included components of ENC. WHO defines SCC as placing the naked baby on the mother's chest with no clothing separating them, in a prone position covered by cloth/blanket [5]. It begins immediately after birth, regardless of the mode of delivery, and is uninterrupted for at least 60 minutes [4].

SCC has numerous benefits for both newborn and maternal health. For newborns, it is a baseline for early initiation of breastfeeding [6–8], regulates body temperature, cardio-respiratory system, physiological function, and blood glucose level [9, 10], helps to transfer good bacteria, boost the maternal to child bonding, reduce baby crying [11]. Whereas for mothers, it shortens the third stage of labour [7, 12, 13], prevents postpartum hemorrhage [14], and reduces maternal stress, and postpartum depression [15]. In general, SCC is a simple, cost-effective, and appropriate method, it has short and long effect importance for both newborns and mothers in particular [16].

Despite the tremendous benefit of SCC, separation of newborns from mothers and putting newborns under warmers are common in many health institutions [17]. A systematic review including literature from 28 countries revealed that the prevalence of SCC varied between 1 to 74% in the LMIC [18]. Evidence suggested that the prevalence of SCC in Sub-Saharan African countries raged between 42% to 45% [19, 20]. In Ethiopia, a few studies were conducted to assess SCC practice. such as a study done by Bedaso et al found that only 28.1% of the mothers practiced SCC in the first hour during the postpartum stay [21]. Similarly, a study conducted

in Gurage Zone found that 35.3% of mothers practiced SCC [22]. This indicates the presence of factors that need further investigation.

Previously, in Ethiopia, studies were conducted to identify factors associated with SCC practice. Accordingly, being urban residence [22], maternal educational status [21], birth weight [22], mother's previous information of SCC [23], number of antenatal care visits (ANC) [21], conducive environment in hospital [23] early initiation of breastfeeding [22, 23], vaginal delivery [23], colostrum feeding and having mothers good knowledge [22] were identified as factors associated with SCC practices. However, these studies were conducted in specific areas of Ethiopia and were not nationally representative. Besides, the use of population-based data is important, as it can provide information regarding the implementation of SCC practices across the country. Moreover, there is a paucity of evidence regarding the spatial distribution of SCC practice of mothers and newborns in Ethiopia. Thus, spatial analysis is important to identify the hot and cold spot areas of mother and newborn SCC practices, not assessed yet. Therefore, this study is aimed to assess the spatial distribution of SCC practice of mothers and newborns and identify its associated factors in Ethiopia using nationally representative data. The result of this study could help policymakers, researchers, program implementers, and other responsible bodies by identifying the factors and cold spot areas of SCC practices.

## Methods

### Data source

The study was done using the Ethiopian Demographic and Health Survey (EDHS 2016) data. It was the fourth survey conducted nationally from January 18 to June 27, 2016. The respondents in EDHS, 2016 were selected using a two-stage stratified sampling technique. In the first stage, 645 enumeration areas (202 in urban areas and 443 in rural areas) were selected. Then in the second stage, 28 households per cluster were selected. The detailed sampling technique was summarized in the full EDHS 2016 report [24]. A total of 10,641 mothers who had birth before the five-year survey were used as a target population. All mothers with live births were used as the study population. All mothers whose births ended with death were excluded from the study. Based on this, a total of 10,417 (weighted samples) mothers with live births were incorporated in the final analysis (Fig 1).

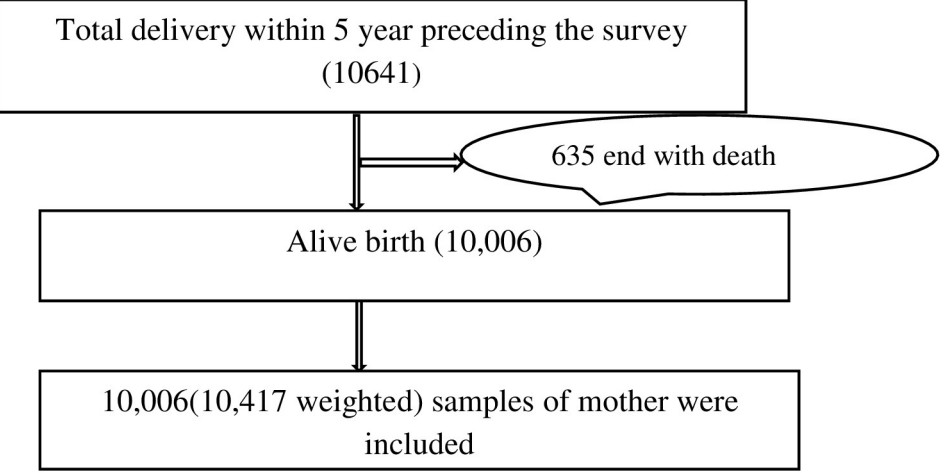

**Fig 1. Schematic presentation of mothers with live births incorporated in the final analysis.**

## Variables of the study

**Dependent variable.**   SCC practice is placing the naked baby on the mother's bare chest, in a prone position covered by a cloth/blanket. Participants were asked whether they practiced or not during birth and their responses were then categorized as yes, coded as "1" and if no coded as "0".

**Independent variables.**   Predictors such as maternal age (years), residence, maternal educational status, maternal working status, number of under-five children, wealth index, media exposure status, region, women participation in making health care decisions, antenatal care visits, place of delivery, sex of the child, birth size, time to breastfeeding initiation, type of birth and parity were independent variables. The media exposure status of mothers was measured using three variables; frequency of watching TV, reading a newspaper, and listening to the radio, and coded as "yes" if an individual was exposed to all or either of the three and "No" if an individual was not exposed to at least one of these. The women's health care decision-making autonomy was assessed as the person who usually decides to obtain healthcare. Which was categorized as women participating in making healthcare decisions and not participating in making healthcare decisions (decided by their husbands/partners).

## Statistical analysis

Data were extracted using SPSS version 21 software and further analysis was done using R version 4.1.3. Data coding, editing, and cleaning were done. Frequencies and percentages were used to summarize data and data presentation was done using tables and Maps. To restore the representativeness of the survey and get reliable statistical estimates, data were weighted for sample probabilities and non-response using the weighting factor provided in the EDHS data (i.e.V005).

## Multilevel analysis

In this study, a two-level hierarchy of data were considered due to the sampling techniques used in the EDHS data (multi-stage stratified cluster). Level one units were individual mothers in households and level two units were enumeration areas (community level). Level one mothers were nested in the households, then the households were nested at the next higher level of enumeration areas (community level). When data have a cluster, subjects within the same cluster may have common characteristics or outcomes, which violates the assumption of independent observations in the classic logistic regression model. Therefore, we used multilevel logistic regression; 1) to take into account the heterogeneity between clusters,2) to account for the intra-cluster correlation of observations in a particular cluster, 3) to examine the cluster (community level) and individual-level variables simultaneously in this nested data. The variance inflation factor was used to explore multicollinearity between independent variables. Both bivariable and multivariable multi-level logistic regression analyses were done. In bivariable two-level binary logistic regression, variables with P-value ≤ 0.25 were a candidate for multivariable multilevel logistic regression analysis. Finally, in multivariable multilevel logistic regression, variables with a p-value of < 0.05 were declared as statistically significant associated factors of SCC practices.

Four models were developed, with the hypothesis of varying intercepts across clusters but fixed coefficients. The first was the null model (Model I) employed without predictor variables. The second model (model II) was fitted for individual-level factors and adjusted to explore its contribution to the variation of SCC practices of mothers and newborns. The third model (Model III) was adjusted for community-level factors and employed to explore its contribution to the variation of SCC practices across the clusters. Finally, the fourth model(Model IV) was

fitted by combining the individual and community-level factors. The model goodness of fit was checked using deviance information criteria (DIC)and Akaike's information criterion (AIC), the model with the lowest value was selected as the best-fitted model.

**Random effects.**   It measures the variation of SCC practices across the clusters using the intraclass correlation coefficient (ICC), median odds ratio (MOR), and proportional change in variance (PCV) statistics. The ICC measures the variation within-cluster and between-cluster differences. The PCV measures the total variation of SCC practices at individual and community-level factors. The MOR measures the median odds ratio of SCC practices at the low-risk cluster (clusters that have better SSC practice during birth) and high-risk cluster(clusters that have low SCC practices) when we select randomly two mothers who have had live birth from two clusters. The formulas for these 3 measurements are as follows:

ICC $= v_i/(v_i + \pi^2/3) \sim \frac{Vi}{Vi+3.29}$, where Vi = between cluster (community) variances and π2 /3 = within-cluster variance.

PCV $= \frac{Vi-Vy}{Vi}$, where Vi = variances of the null model, where Vy = variance of the model with more terms.

MOR $= \exp.[\sqrt{(2\times)} Vz \times 0.6745] \sim \exp.[0.95\sqrt{Vz}]$ where, Vz, = variance at the community (cluster) level

**Spatial analysis.**   We used ArcGIS version 10.3 software to analyze the spatial autocorrelation, spatial interpolation, and hot and cold spot analysis. The SaTscan version 10.0.2 was used to employ SaTScan analysis.

**Spatial autocorrelation.**   The Global Moran's I statistic was used to confirm whether the spatial distribution of SCC practice of mothers and newborns was clustered, dispersed, or random across Ethiopia. Global Moran's I value is close to − 1 indicates dispersed SCC practices whereas Moran's I value closest to + 1 indicates that SCC practice is clustered and "zero" Moran's I value indicates that SCC practice of mother and newborn was randomly distributed across Ethiopia. Moran's I (P-value < 0.05) indicates the presence of spatial autocorrelation [25].

**Hot and cold spot analysis (Getis-OrdGi* statistic).**   The Getis-OrdGi* statistic was done to measure the variation of SCC practice over the study area using GI* statistics. Z-score was used to determine statistical significance clusters. Statistical output with high GI* indicates a "hotspot" whereas low GI* indicates a "cold spot" area of SCC practices [26].

**A spatial SaTscan analysis.**   The Bernoulli-based spatial SaTscan analysis was conducted to identify statistically significant spatial hot spots/clusters area of SCC practices of mother and newborn. While conducting the SaTScan analysis, mothers who have SCC practices with their newborns were used as cases whereas mothers who have no SCC practices with their newborns were considered as controls. The SaTscan scanning window was moved across the study area to identify statistically significant clusters with better SCC practices. The default maximum spatial cluster size of <50% of the population was used, as an upper limit, which allowed both small and large clusters to be detected and ignored clusters that contained more than the maximum limit. Clusters were identified using high log-likelihood ratio (LLR) tests and significant p-values based on Monte Carlo replications. The spatial window with the highest LLR test was considered the most likely cluster of SCC practices.

**Spatial interpolation.**   The spatial interpolation technique was used to predict the SCC practices of mothers and newborns of the unsampled areas in the country based on sampled enumeration areas. The Kriging spatial interpolation method was used to predict SCC practices in unobserved areas of Ethiopia [27].

### Ethical consideration

Since we have used a secondary data analysis that was publically available from the MEASURE DHS program, ethical approval and participant consent were not required as such. We requested the DHS Program and permission was allowed to download to use the data from (https://dhsprogram.com/data/dataset_admin/login_main.cfm.). The requested data were used anonymously and served only for study purposes. The full information about the ethical issue was available in the EDHS-2016 report.

## Results

### Socio-demographic characteristics

A total of 10,417 weighted Samples of mothers who had live births were included. The study found that 21.9% of mothers practiced SCC. More than half (53.2%) of the mothers were aged between 25–34 years. The majority (88.8%) of the mothers were from rural residencies. About 6857(65.8%)of mothers had no formal education. Two-thirds (67.2%) of mothers had no media exposure, such as (radio, TV, and newspaper). Three-fourths (75%) of the mothers had no participation in making health decisions. Only 32% of mothers had four visits to antenatal care. About 7531(72.3%) of mothers had a home birth, whereas three-fourths of the mothers initiated breastfeeding early in the first hours (Table 1).

### Spatial autocorrelation analysis of SCC practice of mother and newborn

The spatial autocorrelation analysis revealed that the spatial distribution of mother and newborn SCC practice was not random in Ethiopia. The Global Moran's I value (0.48, p < 0.0001) indicated that there was significant clustering of mother-to-newborn SCC practices in Ethiopia (Fig 2).

### Hot and cold spot analysis of SSC practice of mother and newborn

In the Getis OrdGi statistical analysis, the significant hot spot areas (have better practices of SCC) were located in Sothern, Northeast, and Easter part of Tigray, Addis Ababa, between the border of South Benishangul Gumz and West Oromia, Dire Dewa, and Harari, whereas significant cold spot areas (have low- practices of SCC) were aggregated in almost all part of Amhara, at the border between East SNNPR(south nation, nationality, and people regions) and southwest Oromia, at the border between East Benishangul Gumz and west Amhara, at the border of West Gambella, at the border between North SNNPR and West Oromia, East SNNPR, East Somali and central, south and west Afar regions (Fig 3).

### Spatial SaTscan analysis of SCC mother and newborn

In SaTScan analysis, the most likely primary, secondary, and tertiary clusters of SCC practice of mothers and newborns were identified (Table 2). The primary SaTScan encloses 63 primary cluster windows located in Addis Ababa centered at 8.593580 North, 39.121293 East, a radius of 67.61 km, with a relative risk (RR) of 2.39 and log-likelihood ratio (LLR) of 116.80 (p <0.001). This indicates that the SCC practice of mothers and newborns was 2.39 times higher among mothers living in the secondary cluster windows as compared to mothers out of the windows. The second SaTScan containing 82 secondary cluster windows was detected in Dire Dewa and Harari centered at 9.543030 North, 42.036260 East with a 32.53 km radius, RR = 2.02, LLR = 110.45 (p <0.001). The SCC practice of mothers and newborns was 2.02 times higher among mothers living in the secondary cluster as compared to mothers living outside the window. The third most likely cluster was detected in the Tigray region at

**Table 1.  Socio-demographic characteristics of study participants.**

| Variable | Category | Weighted frequency (%) | Skin-to-skin contact care done | |
|---|---|---|---|---|
| | | | No (%) | Yes (%) |
| Maternal age (years) | 15–24 | 2320(22.3) | 1709(21.0) | 611(26.8) |
| | 25–34 | 5540(53.2) | 4321(53.1) | 1219(53.5) |
| | ≥35 | 2557(24.5) | 2109(25.9) | 448(19.7) |
| Maternal education status | No formal education | 6857(65.8) | 5741(70.5) | 1116(49.0) |
| | Have formal education | 3560(34.2) | 2398(29.5) | 1162(51.0) |
| Mother Currently working | No | 7592(72.9) | 6049(74.3) | 1543(67.7) |
| | Yes | 2825(27.1) | 2089(25.7) | 736(32.3) |
| Number of under-five children | <4 | 10153(97.5) | 7925(97.4) | 2228(97.8) |
| | ≥4 | 264(2.5) | 213(2.6) | 51(2.2) |
| Wealth index | Poor | 4885(46.9) | 4175(51.3) | 710(31.2) |
| | Medium | 2159(20.7) | 1713(21.0) | 446(19.6) |
| | Rich | 3374(32.4) | 2251(27.7) | 1123(49.3) |
| Exposed to media | No | 7005(67.2) | 5836(71.7) | 1169(51.3) |
| | Yes | 3412(32.8) | 2302(28.3) | 1110(48.7) |
| Women participating in making health care decisions | No | 2608(25.0) | 2115(26.0) | 493(21.6) |
| | Yes | 7809(75.0) | 6024(74.0) | 1785(78.4) |
| Antenatal care visits | No at all | 2660(36.5) | 2382(43.3) | 278(15.6) |
| | 1–3 visits | 2315(31.8) | 1701(30.9) | 614(34.5) |
| | ≥4 visits | 2309(31.7) | 1419(25.8) | 890(49.9) |
| place of delivery | Home | 7531(72.3) | 6832(83.9) | 699(30.7) |
| | Health Institution | 2886(27.7) | 1307(16.1) | 1579(69.3) |
| Sex of child | Male | 5342(51.3) | 4257(52.3) | 1085(47.6) |
| | Female | 5075(48.7) | 3882(47.7) | 1193(52.4) |
| Birth size | Large to average | 3281(31.5) | 2575(31.6) | 706(31.0) |
| | Average | 4442(42.6) | 3397(41.7) | 1045(45.9) |
| | Small to average | 2694(25.9) | 2167(26.6) | 527(23.1) |
| Time to breastfeeding initiation | >1hour | 1782(25.0) | 1426(26.6) | 356(20.3) |
| | With 1hour | 5333(75.0) | 3937(73.4) | 1396(79.7) |
| Type of birth | Single Birth | 10185(97.8) | 7959(97.8) | 2226(97.7) |
| | Twin birth | 232(2.2) | 180(2.2) | 52(2.3) |
| Parity | 1–3 | 4595(44.1) | 3282(40.3) | 1313(57.6) |
| | 4–6 | 3549(34.1) | 2931(36.0) | 618(27.1) |
| | ≥6 | 2273(21.8) | 1925(23.7) | 348(15.3) |
| Variables | Categories | Weighted frequency (%) | Skin-to-skin contact care done | |
| | | | No (%) | Yes (%) |
| Residence | Urban | 1163(11.2) | 593(7.3) | 570(25.0) |
| | Rural | 9254(88.8) | 7546(92.7) | 1708(75.0) |
| Regions | Tigray | 687(6.6) | 385(4.7) | 302(13.3) |
| | Afar | 105(1.0) | 91(1.1) | 14(0.6) |
| | Amhara | 1966(18.9) | 1576(19.4) | 390(17.1) |
| | Oromia | 4571(43.9) | 3705(45.5) | 866(38.0) |
| | Somali | 476(4.6) | 408(5.0) | 476(4.6) |
| | Benishangul Gumz | 113(1.1) | 84(1.0) | 29(1.3) |
| | SNNPR | 2170(20.8) | 1737(21.3) | 433(19.0) |
| | Gambella | 26(0.2) | 17(0.2) | 9(0.4) |
| | Harari | 24(0.2) | 15(0.2) | 9(0.4) |
| | Aldiss Ababa | 236(2.3) | 97(1.2) | 139(6.1) |
| | Dire Dewa | 44(0.4) | 25(0.3) | 19(0.8) |

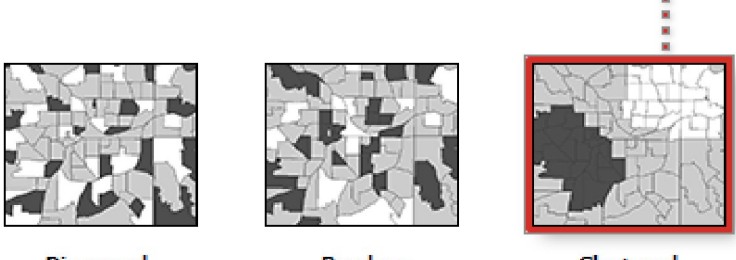

**Moran's Index:** 0.480395
**z-score:** 7.246776
**p-value:** 0.000000

**Fig 2. Spatial autocorrelation of skin-to-skin contact care practices of mothers and newborns in Ethiopia, 2016.**

13.617920 North, 39.332754 East with 55.35 km, RR = 2.32, LLR = 79.20 (p <0.001). The SCC practice of mothers and newborns was 2.32 times higher among mothers who lived in the third cluster than mothers outside the window (Fig 4).

## The spatial interpolation analysis of SCC practices

The spatial kriging interpolation analysis predicted the high and low practice areas of SCC practices. The high and low practice areas of SCC of mothers and newborns were colored white and green, respectively. Northern Amhara, North and South Afar, East Somalia, and between the border of southwest Somali and South Oromia, at the border of west Gambella, at the border between North SNNP and west Oromia regions were predicted as having lower practices of SCC compared to other regions. On the opposite, East Tigray, Addis Ababa, Dire Dewa, and Harari regions were predicated as having better practices of SCC practice (Fig 5).

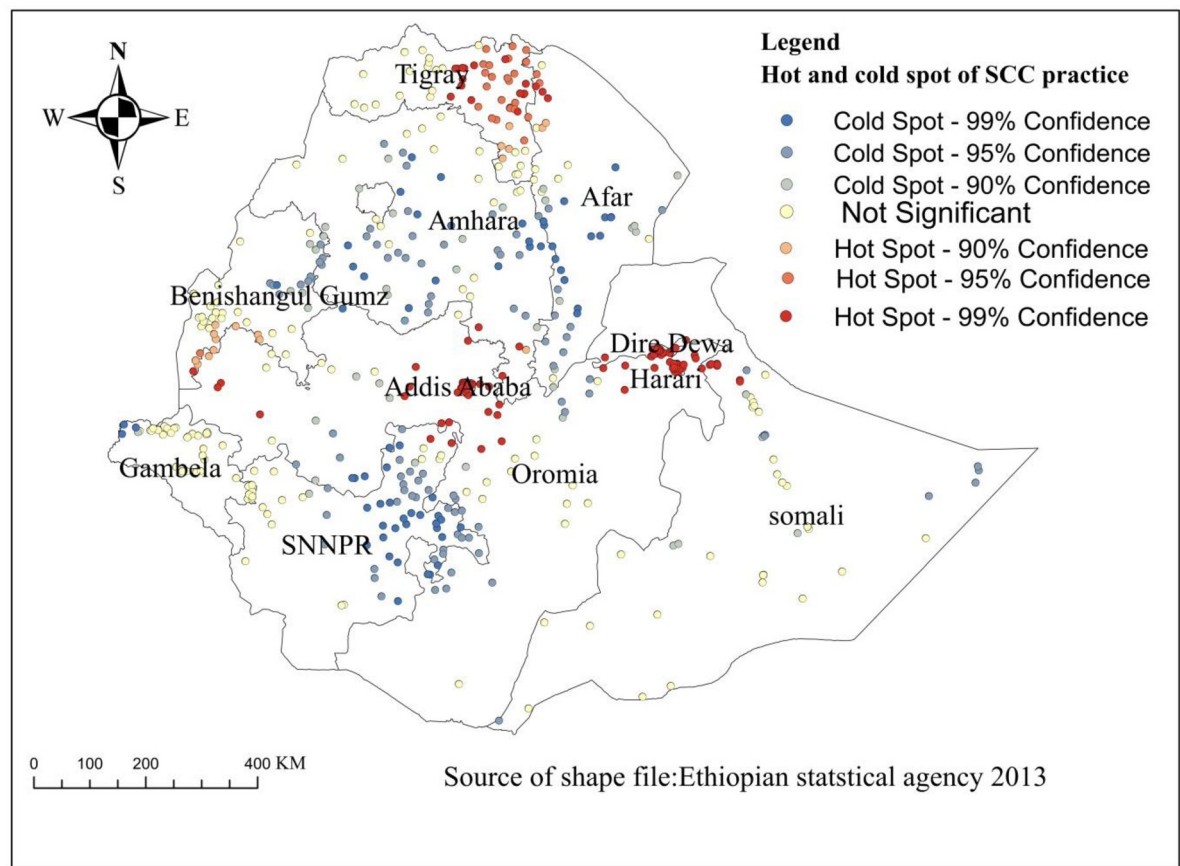

**Fig 3. Hot and cold spot analysis of skin-to-skin contact care practices of mothers and newborns in Ethiopia, 2016.** SCC: skin to skin contact care practices.

**Table 2. SaTscan analysis of significant clusters of skin to skin contact care practices of mothers and newborns, Ethiopia, 2016.**

| Type of cluster | Enumeration areas (clusters) detected | Coordinates/radius | Population | Cases | RR | LLR | P-value |
|---|---|---|---|---|---|---|---|
| First | 290, 149, 125, 40, 353, 83, 236, 252, 303, 211, 475, 330, 261, 539, 451, 402, 524, 90, 61, 264, 155, 560, 225, 509, 428, 19, 293, 302, 110, 287, 639, 159, 247, 15, 414, 582, 59, 195, 635, 645, 305, 153, 608, 314, 170, 487, 464, 145, 147, 108, 100, 31, 626, 144, 369, 107,91, 532, 112, 339, 11, 438, 463 | (8.593580 N, 39.121293 E) / 67.61 km | 509 | 285 | 2.39 | 116.80 | <0.001 |
| Second | 523, 242, 281, 311, 642, 166, 473, 202, 352, 613, 514, 493, 444, 27, 519, 471, 185, 5, 535, 385, 224, 390, 273, 631, 606, 74, 151, 610, 111, 380, 467, 363, 282, 30, 190, 644, 381, 495, 101, 43, 288, 329, 383, 140, 546, 238, 419, 357, 443, 173, 396, 60, 393, 614, 28, 228, 157, 397, 56, 387, 257, 534, 607, 44, 179, 594, 321, 29, 58, 418, 194, 240, 500, 587, 115, 133, 483, 580, 25, 557, 622, 68 | (9.543030 N, 42.036260 E) / 32.53 km | 943 | 436 | 2.02 | 110.45 | <0.001 |
| Third | 430, 94, 237, 550, 220, 605, 623, 384, 355, 129, 424, 538, 99, 226, 298, 575, 604, 160, 196, 341, 579, 421 | (13.617920 N, 39.332754 E) / 55.35 km | 362 | 201 | 2.32 | 79.20 | <0.001 |
| Fourth | 114, 469, 47, 291, 221, 549, 231, 63 | (8.241746 N, 34.605138 E) / 2.93 km | 82 | 58 | 2.86 | 37.96 | <0.001 |
| Fifth | 508, 581, 317, 203, 165, 595, 6, 409, 407 | (10.115443 N, 34.462666 E) / 31.44 km | 136 | 79 | 2.36 | 33.80 | <0.001 |
| Sixth | 349, 70, 304 | (9.528826 N, 35.631217 E) / 26.39 km | 22 | 16 | 2.91 | 11.00 | 0.001 |

RR: relative risk, LLR: log-likelihood ratio

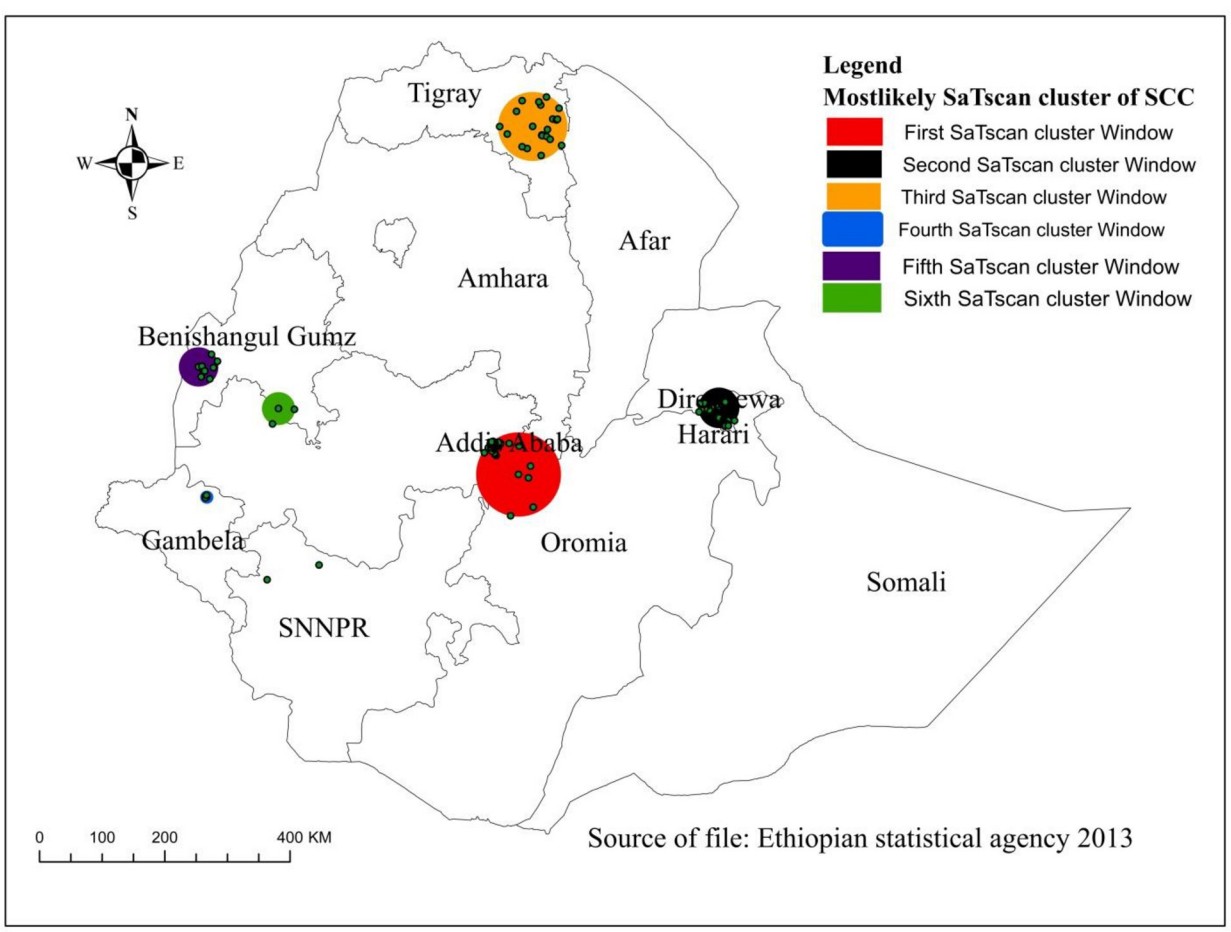

**Fig 4. Spatial SaTscan analysis of skin-to-skin contact care practices of mothers and newborns, Ethiopia, 2016.** SCC: skin to skin contact care practices.

## Random effect and model comparison

Model four with a DIC (5487.5) and AIC (5543.5) value was selected as the best-fitted model to explain the SCC practices of mothers and newborns. The value of ICC in the null model was (0.38), which indicates that 38% of the variability of SCC practice of mothers and newborns was due to differences in clusters. This indicates that the multilevel logistic regression model is the best-fit model to predict the SCC practices of mothers and newborns than the classic logistic regression model. The value of MOR (3.88) in the null model indicates that there was a variation of SCC practices between clusters, which implies that when we randomly select two mothers from two clusters, mothers from a low-risk cluster (clusters have good SCC practice) were 3.88 times more likely practiced SCC than mothers from a high-risk cluster (clusters didn't practice SCC). The higher PCV value in model four (0.798) shows that about 79.8% of the variability of SCC practices of mother and newborn was explained by both the individual-level and community-level factors (Table 3).

## Factors associated with SCC practices

According to the multivariable multilevel logistic regression model, the age of mothers (years), place of delivery, wealth index, media exposure of mothers, antenatal care visits, and time to

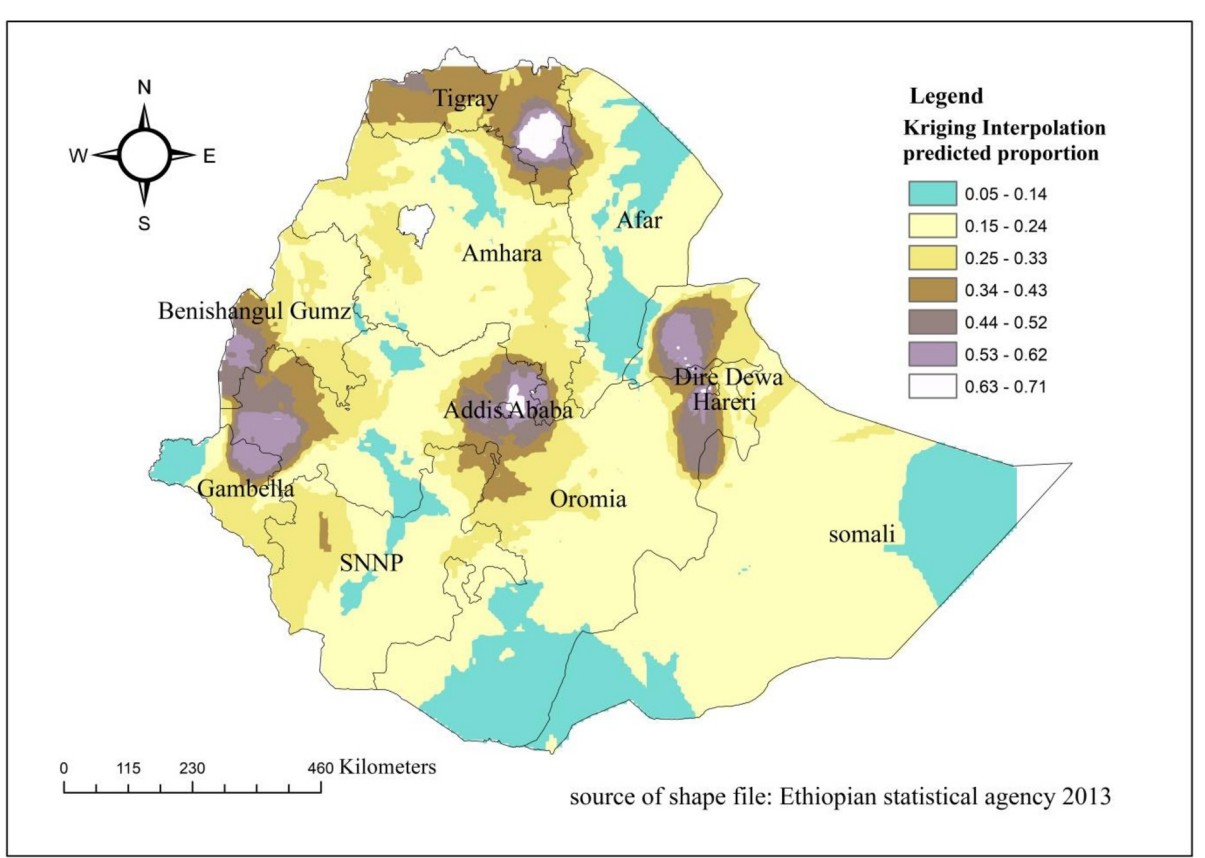

**Fig 5. Spatial interpolation of skin-to-skin contact care practices of mothers and newborns in Ethiopia, 2016.**

breastfeeding initiation were factors associated with SCC practice. Such that, the odds of SCC practices were reduced by 24% (AOR = 0.74, 95%CI: 0.58, 0.96) among mothers whose ages are ≥35 as compared to mothers whose ages are between 15–24 years. The odds of SSC practice were 12.29 times higher (AOR = 12.29, 95%CI: 10.41, 14.54) among mothers who delivered at health facilities as compared to their counterparts. The odds of SCC practices were higher by 75% (AOR = 1.75, 95%CI: 1.49, 2.05) among mothers who initiated breastfeeding within the first hour of birth as compared to mothers who didn't start within the first hour. The odds of SCC practices were higher by 86% (AOR = 1.86, 95%CI: 1.56, 2.29) and 93% (AOR = 1.93, 95CI: 1.56, 2.39) among mothers who have 1–3 and ≥4 ANC visits, respectively, compared to mothers

**Table 3. Random effect and model comparison for predicting factors of skin-to-skin contact care practices of mothers and newborns, Ethiopia, 2016.**

| Random effect | Null | Model II | Model III | Model IV |
|---|---|---|---|---|
| Variance | 2.034 | 0.46 | 0.82 | 0.41 |
| AIC | 9851.3 | 5544 | 9471.2 | 5543.5 |
| DIC | 9847.3 | 5509.7 | 9445.2 | 5487.5 |
| ICC | 0.38 | 0.12 | 0.199 | 0.11 |
| MOR | 3.88 | 1.90 | 2.36 | 1.84 |
| PCV | Ref | 0.77 | 0.59 | 0.798 |

DIC: Deviance information criterion, ICC: intraclass correlation coefficient; MOR: Median odds ratio; PCV: Proportional change in variance.

who didn't have ANC visits. The odds of SCC practices were higher by 20% among mothers who have media exposure (AOR = 1.20, 95%CI: 1.02, 1.41) as compared to mothers who didn't have media exposure. The odds of SCC practices were higher by 29% (AOR = 1.29, 95%CI: 1.05, 1.59) and 38% (AOR = 1.38, 95CI: 1.17, 1.68) among mothers whose wealth index was medium and rich, respectively, as compared to mothers whose wealth index are poor (Table 4).

**Table 4. The multivariable multilevel analysis of factors associated with skin-to-skin contact care practices of mothers and newborns, Ethiopia, 2016.**

| Variable | Category | Null | Model II | Model III | Model IV |
|---|---|---|---|---|---|
| | | | AOR (95%CI) | AOR (95%CI) | AOR (95%CI) |
| Maternal age (years) | 15–24 | | 1 | | 1 |
| | 25–34 | | 0.84(0.71,0.99) | | 0.86 (0.72,1.03) |
| | ≥35 | | 0.72 (0.56, 0.91) | | **0.74 (0.58, 0.96)**\* |
| Maternal education status | No formal education | | 1 | | 1 |
| | Have formal education | | 0.94(0.79, 1.09) | | 0.92 (0.78, 1.08) |
| place of delivery | Home | | 1 | | 1 |
| | Health Institution | | 12.733(10.84,14.99) | | **12.29(10.41, 14.54)**\* |
| Parity | 1–3 | | 1 | | 1 |
| | 4–6 | | 1.062(0.88, 1.27) | | 1.03 (0.85, 1.24) |
| | ≥6 | | 1.32 (1.02,1.70) | | 1.27 (0.98,1.65) |
| Wealth index | Poor | | 1 | | 1 |
| | Medium | | 1.26 (1.02, 1.55) | | **1.29 (1.05, 1.59)**\* |
| | Rich | | 1.38 (1.16,1.66) | | **1.38 (1.17, 1.68)**\* |
| Women participating in making health care decisions | No | | 1 | | 1 |
| | Yes | | 0.92 (0.78,1.08) | | 0.92 (0.79, 1.08) |
| Exposed to media | No | | 1 | | 1 |
| | Yes | | 1.22 (1.04, 1.43) | | **1.20 (1.02,1.41)**\* |
| Antenatal care visits | No at all | | 1 | | 1 |
| | 1–3 visits | | 1.92(1.56, 2.36) | | **1.86 (1.56, 2.29)**\* |
| | ≥4 visits | | 2.06 (1.64,2.50) | | **1.93 (1.56, 2.39)**\* |
| Time to breastfeeding initiation | >1hour | | 1 | | 1 |
| | With 1hour | | 1.78 (1.52,2.07) | | **1.75 (1.49, 2.05)**\* |
| Birth size | Large | | 1 | | 1 |
| | Average | | 1.15 (0.98,1.33) | | 1.14 (0.97, 1.33) |
| | Small | | 0.95 (0.79, 1.14) | | 0.97 (0.81,1.17) |
| Residence | Urban | | | 1 | 1 |
| | Rural | | | 0.21 (0.185, 0.24) | 0.83 (0.66, 1.03) |
| Region | Tigray | | | 1 | 1 |
| | Afar | | | 0.17 (0.13,0.21) | 0.81 (0.57, 1.16) |
| | Amhara | | | 0.34(0.27, 0.42) | 0.63 (0.47,0.86) |
| | Oromia | | | 0.38 (0.32,0.47) | 0.86 (0.66, 1.13) |
| | Somali | | | 0.20(0.16, 0.25) | 0.68 (0.49, 0.93) |
| | Benishangul Gumz | | | 0.53 (0.43,0.65) | 1.24 (0.92,1.66) |
| | SNNPR | | | 0.40 (0.33,0.49) | 0.73 (0.56, 0.95) |
| | Gambela | | | 0.38(0.30, 0.47) | 0.77 (0.56, 1.05) |
| | Harari | | | 0.67 (0.53,0.83) | 0.88 (0.64, 1.21) |
| | Addis Ababa | | | 0.51 (0.39,0.66) | 0.58 (0.43,0.81) |
| | Dire Dewa | | | 0.64 (0.505,0.807) | 0.92 (0.67,1.26) |

1: Reference, OR: Adjusted Odds Ratio, CI: Confidence Interval,

\* = P < 0.05

## Discussion

This study disclosed the spatial variation and determinants of SCC practices of mothers and newborns in Ethiopia using 2016 EDHS data. Accordingly, the SCC practice of mothers and newborns was non-random across the country with Global Moran's I (0.480, p < 0.0001). Besides, place of delivery, wealth index, ANC visits, early initiation of breastfeeding, media exposure, and age of the mother were factors associated with SCC practice.

In hot and cold spot analysis, the significant hot spot areas (have better practices of SCC) were located in Sothern, Northeast, and Easter part of Tigray, Addis Ababa, at the border between South Benishangul Gumz and West Oromia, Dire Dewa, and Harari, whereas the cold spot areas (have low- practices of SCC) were aggregated in almost all part of Amhara, at the border between East SNNPR and Southwest Oromia, at the border between East Benishangul Gumz and West Amhara, at the border of West Gambella, at the border between North SNNPR and West Oromia, East SNNPR, East Somali and Central, South and West Afar regions. The possible justification for the discrepancy might be due to the variation in access to maternal and child health services. This might be because of the difference in the level of awareness and level of education among mothers between the regions. Moreover, there might be a difference in monitoring and evaluation regarding the accomplishment of newborn care packages in these regions.

According to multivariable multilevel logistic regression, the odds of SCC practice were higher among mothers who have media exposure as compared to mothers who didn't have media exposure. The finding was supported by studies conducted elsewhere [20, 28]. The possible justification might be that mothers who are exposed to media might have an awareness regarding the importance of newborn care, including SCC practice; this will increase the willingness of mothers for SCC practice after birth. In addition, this is the fact that media exposure could change the health-seeking behavior of mothers and increase maternal health service utilization [29, 30]. These can open an opportunity for health professionals to provide health education for mothers on the benefits of SCC practices for newborns and mothers.

Consistent with studies conducted elsewhere [20, 21, 28, 31], this study revealed that mothers who have completed ANC visits are associated with a high odds of SCC practices as compared to mothers who have no ANC visits. The possible explanation might be that ANC visits are an opportunity to educate, counsel, and prepare pregnant women for births. Education provided during ANC visits could sensitize women about SSC and its benefits. This will increase adherence and compliance to SSC practices, and finally lead mothers to provide SSC practices to their newborns.

The other striking finding of this study was that mothers who delivered at health facilities are associated with high odds of SCC practices as compared to their counterparts. The finding is consistent with studies conducted elsewhere [20, 28, 31, 32]. It is a fact that in the health facility, there are trained and skilled health professionals, such that they can support mothers to practice SCC. In addition, mothers who are delivered in health facilities possibly are educated about the benefits of SCC practices compared to mothers who have a home birth.

The other observed finding of this study was that mothers from medium to rich household wealth index have higher odds of SCC practice as compared to mothers from poor household wealth index. The finding is supported by studies conducted elsewhere [20, 28]. This was evidenced that mothers with a higher wealth index are more likely educated and utilize maternal and child health services such as ANC, and delivery in health institutions [33, 34].

In this study, mothers who initiate breastfeeding within the first hour are associated with a high odds of SCC practice as compared to mothers who initiate breastfeeding after 1 hour. The result is synonymous with studies [22, 23]. The possible explanation might be that early

initiation of breastfeeding will increase mother-to-child bonding and this will increase the probability of SCC practice. Previous studies confirmed that SCC practices and breastfeeding initiation have a positive association with each other and that the likelihood of breastfeeding initiation is higher among newborns who received SCC practices [7, 8, 35].

Unlike the studies [20, 28, 32], in this study, the age of mothers has a negative association with the SCC practice of mothers and newborns. The study revealed that the odds of SCC practice are lower among mothers aged ≥35 years as compared to mothers aged 15–24 years. The possible justification might be due to the improvement of health education regarding newborn care, girl education, and women's empowerment in making health care decisions [36]. In general, researchers, program implementers, and policymakers should consider the variation and the aforementioned factors to improve the SSC practice of mothers and newborns at the population level.

## Strengths and limitations

The study used nationally representative weighted data and can be generalized at the national level. Besides, the study used an appropriate statistical approach (multilevel mixed analysis) to predict factors of SCC practices. Moreover, the study detected the cold spot areas of SCC practices using spatial analysis, which helps the policymakers to set priorities and plan targeted interventions for the identified cold spot areas. However, the study shares the limitation of a cross-sectional study in that it is impossible to establish the cause-and-effect relationship. Also, data were self-reported and so prone to recall biases despite an attempt taken to minimize recall bias.

## Conclusion

This study concludes that the SCC practice of mother and newborn is not random in Ethiopia. In general, place of delivery, wealth index, ANC visits, early initiation of breastfeeding, media exposure, and age of the mother are factors associated with SCC practice of mothers and newborns Therefore, the implementation of essential newborn care packages should be regularly monitored and evaluated, particularly in the cold spot areas of SCC practices. Besides, media advertising regarding the importance of SCC for mothers and newborns should be scaled up to increase the practices. In addition, strengthening maternal and child health services utilization should be emphasized. Furthermore, special attention to older mothers and those from poor household wealth indexes should be given to increase the practices of SCC.

## Acknowledgments

We would like to thank the Measure DHS program for providing the data set.

## Author Contributions

**Conceptualization:** Zinie Abita.

**Data curation:** Desalegn Girma, Zinie Abita, Gossa Fetene Abebe.

**Formal analysis:** Desalegn Girma, Zinie Abita, Yilkal Negese, Gossa Fetene Abebe.

**Methodology:** Desalegn Girma, Zinie Abita.

**Software:** Desalegn Girma, Zinie Abita, Yilkal Negese, Gossa Fetene Abebe.

**Visualization:** Yilkal Negese.

**Writing – original draft:** Desalegn Girma, Zinie Abita.

**Writing – review & editing:** Desalegn Girma, Zinie Abita, Yilkal Negese, Gossa Fetene Abebe.

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
