## [Decision Letter · Decision Letter 0]

28 Nov 2023

PONE-D-23-22844Spatial variation and determinants of mother and newborn skin-to-skin care practices in Ethiopia: A spatial and multilevel mixed-effect analysis.PLOS ONE

Dear Dr. Girma,

Thank you for submitting your manuscript to PLOS ONE. After careful consideration, we feel that it has merit but does not fully meet PLOS ONE’s publication criteria as it currently stands. Therefore, we invite you to submit a revised version of the manuscript that addresses the points raised during the review process.

**The experts have submitted their response, and some useful suggestions have been given for the improvement. Please see the reviewer's comments especially reviewer 2.**

We look forward to receiving your revised manuscript.

Kind regards,

Muhammad Farooq Umer, PhD Epidemiology and Health Statistics

Academic Editor

PLOS ONE

6. We note that Figures 3,4 and 5 in your submission contain [map/satellite] images which may be copyrighted. All PLOS content is published under the Creative Commons Attribution License (CC BY 4.0), which means that the manuscript, images, and Supporting Information files will be freely available online, and any third party is permitted to access, download, copy, distribute, and use these materials in any way, even commercially, with proper attribution. For these reasons, we cannot publish previously copyrighted maps or satellite images created using proprietary data, such as Google software (Google Maps, Street View, and Earth). For more information, see our copyright guidelines: http://journals.plos.org/plosone/s/licenses-and-copyright.

a. You may seek permission from the original copyright holder of Figures 3,4 and 5 to publish the content specifically under the CC BY 4.0 license. 

Reviewers' comments:

Reviewer's Responses to Questions

**Comments to the Author**

1. Is the manuscript technically sound, and do the data support the conclusions?

Reviewer #1: Yes

Reviewer #2: Yes

2. Has the statistical analysis been performed appropriately and rigorously? 

Reviewer #1: Yes

Reviewer #2: Yes

3. Have the authors made all data underlying the findings in their manuscript fully available?

Reviewer #1: No

Reviewer #2: Yes

4. Is the manuscript presented in an intelligible fashion and written in standard English?

Reviewer #1: Yes

Reviewer #2: Yes

5. Review Comments to the Author

Reviewer #1: This study explored the spatial variation and determinants of mother and newborn skin-to-skin care practices in Ethiopia.. The comments are as follows:

1. There are some language errors need to be revised in the manuscript.

2. The title of Tables 3 and 4 should be revised, do not start with Table-X shows...

Reviewer #2: This manuscript addresses a pertinent public health issue in the context of developing countries and especially in Ethiopia. The manuscript is very well written and analysed using sound statistical methods. Besides meeting other objectives, Interpolation was also done to depict whole country practices based on the sample selected, it was a good effort on part of the authors.

However, there are few points to improve the manuscript which are as follows:

1. Why the most recent data from EDHS 2019 not used

2. Were all the assumptions from the mixed method analysis met with, explain in manuscript in both cases whether these condiions were met with or not

3. How was weighted average of the sampled mothers taken, mention in the methodology

4. Figure 3 needs to mention the footnotes the full form of LLR etc. Also explain how were log likelihood ratio range selected to be depicted in this figure? What is its significance

5. Figure 4 should mention clearly in footnotes about hot and cold spots same as mentioned in lines 229 and 232 of this manuscript. Legend colors for cold and hot spots are not conspicuous, please make them look much better than they currently are.

6. Language/typos/mistakes can be seen in isolated places like Akaike repeated in line 160, lines 129-30 repeated phrase. Although the expression of language is good overall yet it should be given a thorough keen look.

6. PLOS authors have the option to publish the peer review history of their article (what does this mean?). If published, this will include your full peer review and any attached files.

Reviewer #1: No

Reviewer #2: No

---

## [Author Response · Author response to Decision Letter 0]

17 Dec 2023

Reviewer #1 comments and an author response

Comments

Reviewer #1: his study explored the spatial variation and determinants of mother and newborn skin-to-skin care practices in Ethiopia. The comments are as follows:

Authors: dear reviewer, we thank you heartily for your constructive and supportive comments.

Comment #1: There are some languages errors need to be revised in the manuscript.

Authors Response: Thanks very much for your comments, dear reviewer, as per your recommendation, corrective revisions were made in the revised manuscript. 

Comment#2: The title of Tables 3 and 4 should be revised, do not start with Table-X shows.

Authors Response: Thanks very much for your comments, dear reviewer, as per your suggestion; correction was made in the revised manuscript. 

Reviewer #2: This manuscript addresses a pertinent public health issue in the context of developing countries and especially in Ethiopia. The manuscript is very well written and analysed using sound statistical methods. Besides meeting other objectives, Interpolation was also done to depict whole country practices based on the sample selected, it was a good effort on part of the authors.

Authors: dear reviewer, we thank you heartily for your constructive and supportive comments.

Comment 1: Why the most recent data from EDHS 2019 not used

Authors Response: Thanks very much, for your concern, dear reviewer, and the reason why we didn’t use the EDHS 2019 is because it is a mini report and some of the data set was incomplete. For example, here in this study, media exposure was identified as a significant factor of Skin to skin contact care practice of mothers and newborns, but it was not reported in the EDHS 2019 data set.

Comment 2: Were all the assumptions from the mixed method analysis met with, explain in manuscript in both cases whether these conditions were met with or not

Authors Response: Thanks very much, dear reviewer, the comment was accepted; the correction was made in the revised manuscript. It is indicated on pages 7-8, from lines 150-156. 

Comment 3: How was weighted average of the sampled mothers taken, mention in the methodology

Authors Response: Thanks very much, dear reviewer, we accept your suggestion, and correction was made in the revised manuscript, indicated on page 7 from lines 141-144.

Comment 4: Figure 3 needs to mention the footnotes the full form of LLR etc. Also explain how were log likelihood ratio range selected to be depicted in this figure? What is its significance?

Authors Response: Thanks very much for your concern, dear reviewer; we accepted your concern. Dear reviewer, if we are right, your concern is mainly about Fig 4, if so as per your concern, a correction was made in the revised manuscript, please, look at Fig 4. 

Comment 5: Figure 4 should mention clearly in footnotes about hot and cold spots same as mentioned in lines 229 and 232 of this manuscript. 

Authors Response: Thanks very much for your constructive comments, dear reviewer; we accepted your concern. A correction was made in the revised manuscript; please look at the manuscript on page 14, from lines 261-272.

Comment 6: Legend colors for cold and hot spots are not conspicuous; please make them look much better than they currently are.

Authors Response: Thanks very much for your insightful comments, dear reviewer; we accepted your comment. Please, look at the improved legend in Fig 3. 

Comment 7: Language/typos/mistakes can be seen in isolated places like Akaike repeated in line 160, lines 129-30 repeated phrase. Although the expression of language is good overall yet it should be given a thorough keen look.

Authors Response: Thanks very much for your critical insight, dear reviewer, correction was made in the revised manuscript.

---

## [Decision Letter · Decision Letter 1]

3 Jan 2024

Spatial variation and determinants of mother and newborn skin-to-skin contact care practices in Ethiopia: A spatial and multilevel mixed-effect analysis.

PONE-D-23-22844R1

Dear Dr. Girma,

We’re pleased to inform you that your manuscript has been judged scientifically suitable for publication and will be formally accepted for publication once it meets all outstanding technical requirements.

Kind regards,

Muhammad Farooq Umer, PhD Epidemiology and Health Statistics

Academic Editor

PLOS ONE

Additional Editor Comments (optional):

Reviewers' comments:

Reviewer's Responses to Questions

**Comments to the Author**

1. If the authors have adequately addressed your comments raised in a previous round of review and you feel that this manuscript is now acceptable for publication, you may indicate that here to bypass the “Comments to the Author” section, enter your conflict of interest statement in the “Confidential to Editor” section, and submit your "Accept" recommendation.

Reviewer #2: All comments have been addressed

2. Is the manuscript technically sound, and do the data support the conclusions?

Reviewer #2: Yes

3. Has the statistical analysis been performed appropriately and rigorously? 

Reviewer #2: Yes

4. Have the authors made all data underlying the findings in their manuscript fully available?

Reviewer #2: Yes

5. Is the manuscript presented in an intelligible fashion and written in standard English?

Reviewer #2: Yes

6. Review Comments to the Author

Reviewer #2: The queries were addressed satisfactorily by the authors. There are no further comments on this manuscript.

7. PLOS authors have the option to publish the peer review history of their article (what does this mean?). If published, this will include your full peer review and any attached files.

Reviewer #2: No

---

## [Editor Report · Acceptance letter]

12 Feb 2024

PONE-D-23-22844R1 

PLOS ONE

Dear Dr. Girma, 

I'm pleased to inform you that your manuscript has been deemed suitable for publication in PLOS ONE. Congratulations! Your manuscript is now being handed over to our production team.

Kind regards, 

on behalf of

Dr. Muhammad Farooq Umer 

Academic Editor

PLOS ONE